# Transient Receptor Potential Vanilloid 4 (TRPV4) in the Human Carotid Body

**DOI:** 10.3390/ijms262110689

**Published:** 2025-11-03

**Authors:** Yolanda García-Mesa, Elda Alba, Graciela Martínez-Barbero, Iván Suazo, Patricia Cuendias, José Martín-Cruces, Mirian Teulé-Trull, José A. Vega, Olivia García-Suárez, Teresa Cobo

**Affiliations:** 1Grupo SINPOS, Departamento de Morfología y Biología Celular, Universidad de Oviedo, 33003 Oviedo, Spain; garciamyolanda@uniovi.es (Y.G.-M.); gracielamartinezbarbero@gmail.com (G.M.-B.); cuendiaspatrica@uniovi.es (P.C.); pepe3214@gmail.com (J.M.-C.); miriamtaule@gmail.com (M.T.-T.); garciaolivia@uniovi.es (O.G.-S.); 2Instituto de Investigación Sanitaria del Principado de Asturias, ISPA, 33011 Oviedo, Spain; 3Instituto de Neurociencias Vithas, 28010 Madrid, Spain; eldamaselda@gmail.com; 4Servicio de Neurología, Hospital Clínico San Carlos, 28040 Madrid, Spain; 5Facultad de Ciencias de la Salud, Universidad Autónoma de Chile, Santiago 8330015, Chile; ivan.suazo@uatonoma.cl; 6Master ENDORE, Universidad de Santiago de Compostela, 15705 Santiago de Compostela, Spain; 7Departamento de Cirugía y Especialidades Médico-Quirúrgicas, Universidad de Oviedo, 33003 Oviedo, Spain; teresacobo@uniovi.es; 8Instituto Asturiano de Odontología, 33006 Oviedo, Spain

**Keywords:** carotid body, transient receptor potential vanilloid 4, immunofluorescence

## Abstract

TRPV4 (transient receptor potential vanilloid 4) is a non-selective, multifunctional cationic channel that is expressed in numerous cells in the body. It can be activated by temperature, mechanical forces, and chemical and biochemical molecules. Functionally, TRPV4 participates in maintaining osmotic homeostasis, blood pressure, and hypoxic preconditioning. As far as we know, the presence of TRPV4 has never been reported in the carotid body despite the overlap that exists between some biological functions of TRPV4 and the physiology of the carotid body. In the present work, immunofluorescence associated with confocal laser microscopy, associated with quantitative analysis (area occupied by immunofluorescence), has been used to examine the occurrence of TRPV4 in the human carotid body. The results demonstrated the presence of TRPV4 in a subpopulation of chemoreceptor type I cells (approximately 65–68%), a subpopulation of type II supporting cells, and in nerve terminals in the human carotid body. Its function, if any, in this multisensory organ must be demonstrated, but it is in line with the functions attributed to the carotid body.

## 1. Introduction

The carotid body (*glomus caroticum*) is regarded as the most important peripheral chemoreceptor, and it is particularly sensitive to variations in blood levels of oxygen (hypoxemia), carbon dioxide (hypercapnia), and pH (acidosis) [1,2]. However, it should be considered as a polymodal sensor since it also detects temperature, participates in homeostatic control of arterial pressure [3,4], glycemia [5,6,7], in metabolic and hormonal regulation [8,9,10], as well as in the control of sympathetic and parasympathetic activity [11,12]. In addition, some studies indicate that the carotid body senses inflammatory molecules [12,13,14] and therefore could be involved in the inflammatory processes that accompany the physiopathological processes in which the carotid body participate [2,15,16,17].

Structurally the carotid body consists of clusters of chemosensory, or type I, cells, surrounded by supporting, or type II, cells that form the so-called glomic glomeruli, which in turn are grouped into glomic lobules; between the glomerulus and lobules are nerve bundles, arteries, and veins embedded in fibrous tissue that at the periphery form a capsule. Type I cells are a heterogenous population and receives afferent innervation from neurons whose bodies are localized in the petrosal ganglion of the glossopharyngeal nerve. The type II form networks surrounding type I cells and normally lack innervation. On the other hand, nerve fibres that originated in the post-ganglionic neurons of the superior sympathetic ganglion give efferent innervation to the glomic arteries and type I cells. In addition to type I and type II the carotid body contains variable amounts of progenitor cells and neuroblasts (Figure 1) [1].

The physiological and physiopathological ways in which the carotid body participates involve type I cells, type II cells, and afferent and efferent nerves [18,19,20] in the so-called tripartite synapse [21,22,23] all orchestrated by neurotransmitters [2,24,25] and ion channels [2,26,27,28,29,30]. Of particular interest in the physiology of the carotid body are members of the transient receptor potential (TRP) superfamily of ion channels acting alone [26,27,28] or mediating the actions of cytokines [12,13] or leptin [3].

TRPV4 (transient receptor potential vanilloid transient 4) is a non-selective cation channel permeable to Na^+^, Ca^2+^, and Mg^2+^ ions. It consists of four monomers, and each monomer comprises six transmembrane domains with a pore loop in between the fifth and the sixth transmembrane domains [31,32]. TRPV4 is widely expressed in systemic tissues (see [33] for references) and can be activated by temperature [34], mechanical forces [35,36,37,38,39,40], and chemical stimuli (arachidonic acid, 4α-phorbol esters, endocannabinoids, and others) [41,42,43].

TRPV4 plays an important role in maintaining cellular osmotic homeostasis [44], regulating blood pressure to produce both vasodilation and vasoconstriction, and acting as hypoxic preconditioning [45,46,47,48]. Altogether, these data suggest that TRPV4 could be involved in several of the functions of the carotid body [49]. On the other hand, and consistently with the proposed roles for TRPV4, emerging evidence indicates that it is involved in various pathophysiological states, including hypertension, diabetes, and obesity [50,51,52].

Considering the overlapping between the biological functions attributed to TRPV4 and the physiology of the carotid body, we have conducted research to analyze whether TRPV4 is present in this sensory organ. As far as we know, this has never been explored, although TRPV4 has been detected in the petrosal ganglion of the glossopharyngeal nerve that is responsible for the afferent innervation of the carotid body [53]. The present study has been carried out on the human carotid body using immunofluorescence techniques and aims to contribute to the knowledge of the functions of the carotid body.

## 2. Results

The carotid body is structured into glomeruli formed by type I cells, type II cells, blood vessels, and nerves (afferent and efferent). Type II cells are immunofluorescent for the S100 protein (Figure 1a), while type I cells display neuronal-specific enolase (Figure 1b) and synaptophysin (Figure 2b) immunofluorescence. Type II cells are arranged around type I cells, and they are two completely segregated populations for the antigens used as markers (Figure 1c,d).

### Immunolocalization of TRPV4 in the Human Carotid Body

Within the carotid body the glomeruli and lobule are very well defined, and the immunofluorescence for TRPV4 was extremely broad and apparently included all cell types. (Figure 2a). In the co-localization studies, it was observed that the area occupied by type I cells (synaptophysin positive, Figure 2b, and neuron-specific enolase positive, Figure 2e–h) is smaller than that of TRPV4 cells, and in the merge images it was observed that only a subpopulation of type I cells also expresses TRPV4 (Figure 2c–h). The remaining TRPV4 immunofluorescent cells correspond to type II glomic cells, and other cells present in the carotid body such as endothelial cells, cells of the connective septa, progenitor cells, and neuroblasts. The quantitative analysis showed that type I-SYN positive cells or type I-NSE positive cells occupied in the carotid body correspond to an area of approximately 32% and 35%, respectively, whereas that occupied by TRPV4 positive cells correspond to about 68%. Therefore, the percentage of area occupied by TRPV4-positive cells exceeds that of type I cells, suggesting that other cell types of the carotid body, especially type II cells, show immunofluorescence for TRPV4 (occupied approximately 50% of the TRPV4 immunofluorescent area). The results of the quantitative analysis of TRPV4-SYN or TRPV4-NSE mergence show that approximately 65–70% of type I cells express TRPV4 (Table 1 and Figure 2). The small differences in the areas of co-localization of TRPV4 with SYN and NSE may be due to the fact that NSE also marks the nerve fibre terminals.

On the other hand, the presence of TRPV4 in the nerve fibre terminals of the carotid body was investigated. Nerve terminals immunofluorescence for TRPV4 were detected by co-localization immunofluorescence with NSE (Figure 3a–e; it is likely that drop profiles of TRPV4-NSE immunofluorescence merge correspond to transverse sections of nerve terminals) and neurofilament proteins (NFP; Figure 3g,h). Based on the quantitative analysis, about 90% of the nerve fibre terminals in the human carotid body are TRPV4 positive (Table 1 and Figure 2).

Therefore, in the human carotid body, TRPV4 is present in a subpopulation of type I cells and in most of the nerves that supply it, but also in uncharacterized cells that mainly correspond to type II cells.

## 3. Discussion

TRPV4 is a multifunctional non-selective cation channel belonging to the vanilloid family of the transient receptor potential (TRP) ion channels superfamily (see [41,53]). This ion channel has been detected with different techniques in most tissues, such as the smooth muscle cells, vasculature, lungs, brain, heart, kidneys, salivary glands, liver, bladder, trachea, skin, bone, spleen, testicle, dorsal root, and cranial nerve sensory ganglia [33,46,53,54,55,56,57,58,59]. As far as we know, the presence of TRPV4 in the normal human carotid body is demonstrated now for the first time. No data is available about TRPV4 in other peripheral chemoreceptors, whereas it has been found in peripheral baroreceptors [45]. However, other members of the TRP ion channel superfamily have been detected in the carotid body. Buniel and co-workers [26] demonstrated the presence of members of the canonical family of TRPs (TRPC3, TRPC4, TRPC5, TRPC6, and TRPC7) in different neuronal subpopulations of the rat petrosal ganglion that gives afferent innervation to the carotid body, although only TRPC1 and TRPC4 were localized in the afferent nerve terminals of the carotid body. The role of these channels in the chemosensory pathway of the carotid body is yet to be established. One member of the vanilloid TRP family, TRPV1, mediates the sensing of the carotid body to Th2 cytokines and lysophosphatidic acid [12,13], and a member of the melastatin TRP family, TRPM7, is involved in the regulation of obesity-induced hypertension mediated by leptin [3,8]. However, the role of the channels of the different families of TRP channels is far from being fully known [27].

In our hands, specific immunofluorescence for TRPV4, was detected in a subpopulation of type I and in nerve terminals, and presumably also in other cells including type II cells, blood vessel cells, progenitor cells, neuroblasts, and cells of the connective septa. There is evidence that it is present in the satellite glial cells [60] and neurons [61] of the spinal ganglia, as well as in the complex peripheral sensory ganglia that innervates different organs of the carotid body area [54]. So, the occurrence of TRPV4 in the nerve terminals within the carotid body is consistent with those data. The type I carotid body cells and the peripheral sensory ganglia are both derivatives of the neural crest, and type I cells and neurons as well as type II cells and satellite glial cells share common antigens [62]. Experiments are in progress in our laboratory to determine which TRP, ASIC, and PIEZO ion channels are present in type II cells of the human carotid body, and which of them are present in cells of the carotid body other than type I and type II.

Because of the wide spectrum of functions of TRPV4 it is difficult to assign it a single role in the carotid body. However, TRPV4 presumably is related to the functions of this multisensor. The carotid body detects variations in acidosis and TRPV4 is an acidosis detector [63], most likely in association with ASIC channels [29,64]. TRPV4 is also involved in the control of glycemia [50,65], metabolic, and hormonal regulation [66,67], such as the carotid body [4,5,68].

But surely the most obvious role of TRPV4 in the carotid body is as mechanosensor and osmosensor [53,69,70]. In the carotid body, a subpopulation of type I cells, and surely most of type II cells express TRPV4. Interestingly, type I cells express PIEZO2 via mechanosensor–mechanotransducer ion channels, and type II cells express both PIEZO1 and PIEZO2 [30]. Also acid-sensing ion channels which participate in mechano detection were found in type I and type II cells in the human carotid body [29,71].

In view of the present results, it can be hypothesized that the carotid body presumably plays a role in regulating various physiological processes in both normal and pathological conditions, such as many metabolic or cardiovascular diseases and hypertension [48,72,73,74,75,76,77]. However, many more studies are needed to definitively establish the role of TRPV4 in the carotid body.

## 4. Materials and Methods

Tissue samples of the carotid body were obtained during removal of organs for transplantation in subjects who died in traffic accidents (Hospital Universitario Central de Asturias, Oviedo, Spain). There were 8 pieces (from 5 males and 3 females), with ages ranging between 38 and 68 years. The petrosal ganglia (n = 4) and the sympathetic superior ganglion (n = 4) were dissected out and included in the study. The pieces were cleaned in 4 °C saline solution; fixed in 10% formaldehyde in 0.1 M phosphate-buffered saline, pH 7.4, for 24 h at 4 °C; dehydrated; and routinely embedded in paraffin.

These materials were obtained in compliance with the Spanish Law and the guidelines of the Helsinki Declaration II. Sections of the same material had been used in previous studies on the carotid body [16,17]. This material, which was used for research purposes, is deposited in the Department of Morphology and Cell Biology of the University of Oviedo, National Registry of Biobanks (Collections Section, Ref. C-0001627), created and authorized by the Ministry of Economy and Competitiveness of the Government of Spain on 30 November 2012.

### 4.1. Double Immunofluorescence

Tissue samples were cut 10 μm thick and the sections were mounted on gelatin-coated slides. Sections were then indirectly immunohistochemically stained with peroxidase–anti-peroxidase, rehydrated, and rinsed in 0.05 M Tris-HCl buffer (pH 7.5) containing 0.1% bovine serum albumin and 0.1% Triton X-100. The sections were processed for the simultaneous detection of TRPV4 together with neuronal (neurofilament protein, NFP) type I cell (neuron-specific enolase, NSE; synaptophysin, SYN) and type II cell (S100 protein, S100P) markers [29,30,78,79,80] (Table 2. It should be noted that type I glomus cells also express NSE. The antibody against TRPV4 was raised in rabbit anti-human TRPV4 (manufacturer’s notice).

The sections were incubated overnight at 4 °C, in a humid chamber, with a 1:1 (*v*/*v*) mixture of polyclonal antibodies against TRPV4 and monoclonal antibodies against NFP, NSE, S100P, or SYN. After rinsing, the sections were sequentially treated, in a dark and humid chamber, for 1 h at room temperature, with a goat anti-rabbit antibody conjugated to Alexa Fluor 488 (Serotec™, Oxford, UK; 1:1000 dilution) and a donkey anti-mouse antibody conjugated to Cy3 (Jackson-ImmunoResearch™, Baltimore, MD, USA; 1:50 dilution). Finally, cell nuclei were counterstained with DAPI (10 ng/mL). Immunofluorescence was examined using a Leica DMR-XA automated fluorescence microscope equipped with Leica confocal software, version 2.5 (Leica Microsystems, Heidelberg GmbH, Heidelberg, Germany). For controls, some sections were processed following the same protocol, but the primary antibodies were either omitted or replaced with non-immune rabbit or mouse sera.

### 4.2. Quantitative Analysis

A quantitative image analysis was carried out in the carotid body which was processed for the immunodetection of TRPV4 using an automatic image analysis system (Quantimet 550, Leica, Wetzlar, Germany, QWIN Program).

Measurements were made in 10 randomly selected fields per section (5 mm^2^), with five sections for glomus 100 µm apart (total 400 fields). The immunoreactive area for SYN or NSE was considered to be 100% of the area of type I cells; the area occupied by the merging of SYN + TRPV4 immunoreactivities or NSE + TRPV4 were considered as the area of type I cells with immunoreaction for TRPV4. The results are expressed as the values of the mean ± standard error. Also, in 10 randomly selected fields per section (5 mm^2^), with five sections for glomus, 100 µm apart (total 400 fields), the density of nerve terminals innervating the carotid body was based on the detection of NFP immunofluorescence; the area of immunoreactive NFP positive nerve terminals was considered to be 100% of the area occupied by the nerve terminals; and the area occupied by the merging of NFP + TRPV4 was considered to be the area of nerves with immunoreaction for TRPV4. The results are expressed as the area occupied by nerve terminals/mm^2^.

All experiments were performed in duplicate, and the results in both cases were homogeneous.

## Data Availability

The original contributions presented in this study are included in the article. Further inquiries can be directed to the corresponding author.

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
