# Peer review of "Transient Receptor Potential Vanilloid 4 (TRPV4) in the Human Carotid Body"

_ijms, 2025, doi:10.3390/ijms262110689_

Round 1
Reviewer 1 Report
Comments and Suggestions for Authors
This is an interesting study showing the presence of TRPV4 in the human carotid body for the first time. The study's aim is clear, and the findings are relevant to the field. The use of immunofluorescence and quantitative analysis provides compelling evidence for TRPV4's localization in a subpopulation of type I cells, type II cells, and nerve terminals. I have a few suggestions that should be considered before publication.
Comments:
- Introduction and Discussion: These sections should be expanded by including and discussing several studies demonstrating the role of many subtypes of TRP channels within the carotid body (PMID: 33180962, PMID: 30279412, PMID: 33993722, PMID: 31545149, PMID: 12900933).
- Introduction, lines 44-47: Besides temperature, recent studies have demonstrated that the carotid bodies are important sensors of inflammatory mediators (PMID: 33180962, PMID: 36389846, PMID: 35339628). Please include.
- Discussion, lines 147-148: “TRPV4 is also involved in the control of glycemia [36,51], metabolic and hormonal regulation [52,53] such as the carotid body [2,3,54]”. This sentence is not clear. Please rephrase.
- Discussion, line 150-153: “But surely the most obvious role of TRPV4 in the carotid body is as mechanosensor and osmosensor [55,56]”. In the carotid body, a large percentage of type I and type II cells express TRPV4, as well as mechanosensor-mechanotransducer ion cales such as PIEZO2 [17] and pannexin 1 channels (M Anache et al., unpublished) that are also opened by mechanical factors”. How can you be sure on the role of TRPV4 channels in the carotid body? Also, please do not refer to unpublished and therefore not peer-reviewed data.
Comments on the Quality of English Language
Can be rechecked for some typos and grammar mistakes.
Author Response
REVIEWER 1
The authors gratefully acknowledge the anonymous Reviewer for the constructive criticisms.
First of all, the authors would like to apologize to the Reviewer since by mistake the non-final version of the manuscript was submitted and uploaded to the IJMS platform. For this reason, some judgments appear unclear in the original version. Please excuse the inconvenience that has been caused by an error that I, JA Vega, take full responsibility.
According to the Reviewer suggestions, the following changes were introduced in the manuscript (labelled in red):
This is an interesting study showing the presence of TRPV4 in the human carotid body for the first time. The study's aim is clear, and the findings are relevant to the field. The use of immunofluorescence and quantitative analysis provides compelling evidence for TRPV4's localization in a subpopulation of type I cells, type II cells, and nerve terminals. I have a few suggestions that should be considered before publication.
Comments:
- Introduction and Discussion: These sections should be expanded by including and discussing several studies demonstrating the role of many subtypes of TRP channels within the carotid body (PMID: 33180962, PMID: 30279412, PMID: 33993722, PMID: 31545149, PMID: 12900933).
Thank you very much for the suggestion; references to some studies that have demonstrated the presence of TRP superfamily channels in the carotid body and their role in this organ have been included.
- Introduction, lines 44-47: Besides temperature, recent studies have demonstrated that carotid bodies are important sensors of inflammatory mediators (PMID: 33180962, PMID: 36389846, PMID: 35339628). Please include.
Again, thank you very much for the very appropriate suggestion that has been introduced in the revised version of the manuscript.
- Discussion, lines 147-148: “TRPV4 is also involved in the control of glycemia [36,51], metabolic and hormonal regulation [52,53] such as the carotid body [2,3,54]”. This sentence is not clear. Please rephrase.
That paragraph has been redrafted.
- Discussion, line 150-153: “But surely the most obvious role of TRPV4 in the carotid body is as mechanosensor and osmosensor [55,56]”. In the carotid body, a large percentage of type I and type II cells express TRPV4, as well as mechanosensor-mechanotransducer ion cales such as PIEZO2 [17] and pannexin 1 channels (M Anache et al., unpublished) that are also opened by mechanical factors”. How can you be sure on the role of TRPV4 channels in the carotid body? Also, please do not refer to unpublished and therefore not peer-reviewed data.
Thank you very much for both suggestions; In the revised version of the manuscript, this sentence has been clarified.
Reviewer 2 Report
Comments and Suggestions for Authors
The paper may be interesting for some readers, but seems to be written hastly.
- The most serious concern is that the authors basically re-write their previous paper (doi.org/10.3390/biom15030386) but using different antibodies. Do you plan to publish one paper each year with the same samples, but different antibodies? At least the methods should not be repeated word by word.
- Abstract+Introduction: Why is it important to study whether it's possible to detect TRPV4 presence in the carotid body? The paper is not comparing the TRPV4 stain with other organs and other TRPV or related receptor. So what, if I have an image of a western blotting bands of one protein in some specific organ/tissue/cells type, I should just publish it right away? Or detect a specific metabolite? You probably could’ve, if the paper is methodological one, but then it should include some other samples where your staining is different in different tissues or different physiological/pathological conditions
- And "human" as a keyword?
- There are several papers showing immunological staining of TRPV4 in various cells and organs (including carotid arteries and ganglion in Ref. 39, but yes, the “carotid body” itself probably wasn’t analyzed). So, why do you think that detection of a single protein in carotid body is required “to contribute to the knowledge of the functions of the carotid body”? Again, please expand Introduction, Discussion and add Conclusion to show “why” was it done.
- Unify mentioning type I and type II cells. Why not just neurons and glia? Sometimes type I are “chemoreceptor cells”, sometimes it’s “glial”
- Please try to determine these stained “populations”. The percentages themselves are not clear. But if the main idea is that “TRPV4 is mostly in neurons, not in glia cells”, try chi-square test or some other to confirm that and state this clearly everywhere. Now your results just show that there are just some cells with TRPV4 with undefined common phenotype.
- A scheme required to show how to access "the petrosal ganglia (n = 4) and the sympathetic superior ganglion (n = 4)" in the neck. It seems that "carotid body" is a separate structure, not comprising the previous two though.
- Why NSE a S100P with SYN are considered markers of neurons and glial cells? Citations are required.
- Blue is DAPI or what?
- What software is used to make a diagram. The diagram is also hard to understand (lacks legend, the percentages in brackets are higher than without, glial cells are not shown, SYN+ cells are not shown in Figures, etc.). The scheme and caption in your previous paper is much more clear, but you should try even better to write everything, what each color means.
- Author contribution statement: next time please read your manuscript before uploading to see if your formatting is ok.
So, in the end, it’s an interesting, more like methodological, paper, but please make it look less like your previous ones, clearly determine why is it important to detect a single protein in a single organ in “normal” state.
Comments on the Quality of English LanguageThe authors should have read the manuscript carefully before the submission.
What is even "chemical and biochemical molecules"? Other phrases that sound weird the authors should find themselves.
Author Response
REVIEWER 2
The authors gratefully acknowledge the anonymous Reviewer for the constructive criticisms.
First of all, the authors would like to apologize to the Reviewer since by mistake the non-final version of the manuscript was submitted and uploaded to the IJMS platform. For this reason, some judgments appear unclear in the original version that you evaluated. Consequently, some of the questions you raised were already resolved in advance and are reflected in the current manuscript. Please excuse the inconvenience that has been caused by an error that I, JA Vega, take full responsibility.
According to the Reviewer suggestions, the following changes were introduced in the manuscript:
The most serious concern is that the authors basically re-write their previous paper (doi.org/10.3390/biom15030386) but using different antibodies. Do you plan to publish one paper each year with the same samples, but different antibodies? At least the methods should not be repeated word by word.
Thanks for the suggestion. The assessment he makes is true. The material we have, although there are few samples, is almost unique because it has been obtained from organ donors during the extraction process. As it is human material, it is difficult to do more experimentation than structural and morphological experimentation. Honestly, this is the last study on the carotid body that we can carry out because we do not have more material and the present one was limited to TRPV4 because there was not enough material available to study others. We have tried to change the section on material and methods, but honestly there are parts that are impossible to modify (origin of the material, ethical data, antigen detection process).
Abstract+Introduction: Why is it important to study whether it's possible to detect TRPV4 presence in the carotid body? The paper is not comparing the TRPV4 stain with other organs and other TRPV or related receptor. So what, if I have an image of a western blotting bands of one protein in some specific organ/tissue/cells type, I should just publish it right away? Or detect a specific metabolite? You probably could’ve, if the paper is methodological one, but then it should include some other samples where your staining is different in different tissues or different physiological/pathological conditions
Honesty in research has always been the guide of my work. If TRPV4 was studied in the human carotid body, it was mainly for two reasons: a) it was suggested by one of the reviewers of our previous work; 2) most of the functions of TRPV4 are related to those of the carotid body. I believe that this fact is clearly set out in the introduction.
And "human" as a keyword?
It was deleted
There are several papers showing immunological staining of TRPV4 in various cells and organs (including carotid arteries and ganglion in Ref. 39, but yes, the “carotid body” itself probably wasn’t analyzed). So, why do you think that detection of a single protein in carotid body is required “to contribute to the knowledge of the functions of the carotid body”? Again, please expand Introduction, Discussion and add Conclusion to show “why” was it done.
All sections of the manuscript are different from those you originally read, and surely in this version those aspects are considered.
Unify mentioning type I and type II cells. Why not just neurons and glia? Sometimes type I are “chemoreceptor cells”, sometimes it’s “glial”
Thanks for the suggestion. It has been used throughout the text as type I and type II cells. Only in the Abstract and in the first two lines of Results are other terms used because the structure of the human carotid body is not detailed in the manuscript.
We have not used the terms of neurons and glia because they are neither neurons nor peripheral glia; They are closely related by their embryonic origin and the fact that they share some antigens but are different cell populations. In the sensory process, the cells of the carotid body, especially type I, are the ones that detect variations in oxygen, carbon oxide, pH and transmit them to the nerve terminals of the afferent fibers. Axons are neurons while type I cells are the sensors. And the same could be said of type cells as they are involved in the tripartite synapse of the carotid body. This concept is explained in the Introduction.
Please try to determine these stained “populations”. The percentages themselves are not clear. But if the main idea is that “TRPV4 is mostly in neurons, not in glia cells”, try chi-square test or some other to confirm that and state this clearly everywhere. Now your results just show that there are just some cells with TRPV4 with undefined common phenotype.
I believe that all these aspects are in the new version of the manuscript and that as I mentioned at the beginning you did not receive the correct version originally. Thank you for your patience and understanding.
Regarding the typing of the "other cells" of the carotid body TRPV4 positive, we did not obtain good results with the antibodies used to mark type II cells and for that reason it was not included in the study.
A scheme required to show how to access "the petrosal ganglia (n = 4) and the sympathetic superior ganglion (n = 4)" in the neck. It seems that "carotid body" is a separate structure, not comprising the previous two though.
In some animal species they are quite close structures, but in humans are completely different things and they are found in different areas: the carotid body at the bifurcation of the common carotid artery (with some anatomical variability), the petrosal ganglion is usually found inside or at the exit of the jugular foramen (which means at least 4-6 cm above the carotid body depending on the length of the neck) . And with respect to the superior cervical sympathetic ganglion, although it is approximately at the height of the carotid bifurcation, it is very posterior, in the retrostyloid space, also far from the carotid body.
For all these reasons, I consider it unnecessary to include a scheme to locate these formations in humans since they can be found in all anatomy manuals.
On the other hand, since 1998 I have been a full professor of neuroanatomy, I know the human peripheral nervous system well and the dissection and extraction of the pieces used have been carried out myself.
Why NSE a S100P with SYN are considered markers of neurons and glial cells? Citations are required.
Thanks for the suggestion. They are well-known markers of type I and type II cells, and we have added the appropriate references.
What software is used to make a diagram. The diagram is also hard to understand (lacks legend, the percentages in brackets are higher than without, glial cells are not shown, SYN+ cells are not shown in Figures, etc.). The scheme and caption in your previous paper is much more clear, but you should try even better to write everything, what each color means.
No software was used to make the scheme; I personally did it "freehand". In the current version of the scheme it will surely be easier to understand.
I reiterate my apologies for the error made in submitting the inappropriate version of the manuscript. In the version I am submitting now, your suggestions and those of the other reviewers have been considered, which specifically required you to introduce more information about the ion channels of the TRP superfamily.
Round 2
Reviewer 2 Report
Comments and Suggestions for Authors
Yes, this looks a bit cleaner. But some scientific questions still remain unanswered. Also “human” is still a keyword.
I wonder if this is also an incorrect version. And I think all authors, not only the corresponding one, should read the manuscript they participated in to find other typos, bad grammar, methodological errors, discrepancies in cell naming, and other errors.
I got the idea that the material is unique. But please add the information (at least to Introduction), that the material is that you just happen to have an access to that material (cite your previous papers). And not that you just decide to study carotid body, because someone suggested the receptor should be involved in its functioning.
Other remarks include:
- The concept of two types of cells is not fully explained in the Introduction; so, you could’ve added your reply to my remarks to the manuscript itself. In the Results sections the cells are named however “neuron-like” and “glia-like”, so why not continue this.
- Are nerve terminals parts of neurons or type I cells? How are they defined in Fig. 2 (SYN–, NSE+ or just the size of “spots” is different)? If they are separate from type I and type II cells. Could you please add some more labels and rearrange Figs. 2 and 3? Also, the yellow arrows in Figs. are not defined. And what do “nerve profiles” (same as terminals?) and blue color (nuclei?) mean?
- So, no co-localization with S100P was tested? Doesn’t it mean that maybe all type II cells should be stained with TRPV4, which will explain all the green spots in merged Figs. 2 and 3? If there’re no more cells other than type I and II ones, is it possible to calculate the percentage of TRPV-positive type II cells? If there are other cells, than “specific immunofluorescence for TRPV4, was detected in a subpopulation of type I and in nerve fibre terminals, and presumably also in other cells including type II cells” requires some confirmation. But, I believe, 90% of nerve terminals stained with TRPV4 is more than 20% of type I cells and you could rewrite your paper a bit to better indicate the importance of TRPV4 in terminals, whatever this is (just some axons?).
- I still don’t get the diagram. The percentages below the figures mean percentages of all cells or percentages of cells of specific type? Please expand the description.
I doubt that all readers saw the “anatomy manuals” required, please cite. Or why not make another diagram? If “The petrosal ganglia (n = 4) and the sympathetic superior ganglion (n = 4)” are separate structures from carotid body (as it seems from your reply), why were they “included in the study”? And if all these 8 samples were treated as “carotid bodies”, could you clarify how exactly the results were merged (400 images means 10 images x 5 sections x 8 samples, correct?)
Author Response
REVIEWER 2
I wonder if this is also an incorrect version. And I think all authors, not only the corresponding one, should read the manuscript they participated in to find other typos, bad grammar, methodological errors, discrepancies in cell naming, and other errors.
I got the idea that the material is unique. But please add the information (at least to Introduction), that the material is that you just happen to have an access to that material (cite your previous papers). And not that you just decide to study carotid body, because someone suggested the receptor should be involved in its functioning.
Thank you very much for the criticisms of our work which, without a doubt, are born from scientific rigor and with the intention that the manuscript be improved. Our two previous works are already referenced in the manuscript (references 29 and 30) in the context that, in our opinion, they should be.
Other remarks include:
- The concept of two types of cells is not fully explained in the Introduction; so, you could’ve added your reply to my remarks to the manuscript itself. In the Results sections the cells are named however “neuron-like” and “glia-like”, so why not continue this.
Thanks for the suggestion. The terms neuron-like and glial-like have been eliminated throughout the text; and only in lines 50 and 51 of the manuscript are the terms chemoreceptor and support cells used only once to refer to type I and type II cells, respectively. A paragraph has been added in the Introduction (lines 50 to 58) that surely clarifies many of the issues you raise in your criticisms.
Neither neuron-like nor glial-like are used because they are not suitable: the cells of the carotid body are well differentiated even though they resemble neurons or glia. Because from there it would be necessary to explain what type of neurons or glia they resemble. In addition, there are clear differences in their embryology. Type I cells are derived from the neural crest with a neuronal lineage derived from the superior cervical ganglion. The type II sustentacular cells of the carotid body (Figure 2B) form networks of supporting cells. They are derived from mesenchymal neural crest cells and are of glial cell lineage (Kameda, 2020).
- Are nerve terminals parts of neurons or type I cells? How are they defined in Fig. 2 (SYN–, NSE+ or just the size of “spots” is different)? If they are separate from type I and type II cells. Could you please add some more labels and rearrange Figs. 2 and 3? Also, the yellow arrows in Figs. are not defined. And what do “nerve profiles” (same as terminals?) and blue color (nuclei?) mean?
Thank you for the comments. As you are well aware, the concept of "nerve terminal" or "nerve fibre terminal" or "nerve profile" is used in neuroanatomy to refer to nerve fibers that are found in a tissue without being able to trace their entire path, and that cannot be said (as in this case) if they are part of afferent or efferent neurons (for example in Nephew et al. The carotid body: a physiologically relevant germinal niche in the adult peripheral nervous system. Cell Mol Life Sci. 2019; 76:1027-1039. doi: 10.1007/s00018-018-2975-9). In this manuscript it refers to the nerves inside the carotid body. Therefore, this concept refers to neuron axons, type I cells do not have axons.
To avoid confusion or misunderstandings in the revised text, only "nerve terminal" is used.
The yellow arrows in Figure 2 have been identified 2.
In all the figures it has been added that the blue fluorescence corresponds to the DAPI.
- So, no co-localization with S100P was tested? Doesn’t it mean that maybe all type II cells should be stained with TRPV4, which will explain all the green spots in merged Figs. 2 and 3? If there’re no more cells other than type I and II ones, is it possible to calculate the percentage of TRPV-positive type II cells? If there are other cells, than “specific immunofluorescence for TRPV4, was detected in a subpopulation of type I and in nerve fibre terminals, and presumably also in other cells including type II cells” requires some confirmation. But, I believe, 90% of nerve terminals stained with TRPV4 is more than 20% of type I cells and you could rewrite your paper a bit to better indicate the importance of TRPV4 in terminals, whatever this is (just some axons?).
We believe that these questions are already clarified with the previous answers. Nerve terminals were quantified by immunofluorescence of NFP + TRPV4 while type I cells with NSE + TRPV4 and SYN + TRPV4.
- I still don’t get the diagram. The percentages below the figures mean percentages of all cells or percentages of cells of specific type? Please expand the description.
Thank you for your appreciation. In the manuscript, and it is detailed in the Material and Methods section, there is never any mention of percentages of cells or number of cells, but of the area occupied by the immunoreaction for specific markers of type I, type II cells or nerve terminals.
- I doubt that all readers saw the “anatomy manuals” required, please cite. Or why not make another diagram? If “The petrosal ganglia (n = 4) and the sympathetic superior ganglion (n = 4)” are separate structures from carotid body (as it seems from your reply), why were they “included in the study”?
Thanks for the criticism and suggestions. At your request, a diagram has been included where the anatomical location of the carotid body, the petrosal ganglion and the upper cervical sympathetic ganglion is observed in humans.
As for why the ganglia are included in the study, the answer is simple: because the petrosal ganglion is where the somas of the afferent neurons to the carotid body are located, and the superior cervical sympathetic the somas of the efferent neurons of the carotid body. The paragraph added to the introduction clarifies this.
- And if all these 8 samples were treated as “carotid bodies”, could you clarify how exactly the results were merged (400 images means 10 images x 5 sections x 8 samples)
Thanks for the observation. They were not treated as carotid bodies, they are sections of carotid body, and it means exactly that: 10 fields in 5 sections (separated by 100 μm from each other) of each of the samples.
Round 3
Reviewer 2 Report
Comments and Suggestions for Authors
Well, I think you've clarified most of it, although I would like to see the answers within the new version.
Unfortunately, most of interested readers, including me, will not be well aware of terminology used in neuronal development biology, such as "nerve profile", so yes, please try to unify the terms throughout your next manuscripts.
The only thing that is left is co-localization with S100 to indicate the presence of TRPV in type II cells. If "The remaining TRPV4 immunofluorescent cells correspond to type II glomic cells, although the presence of TRPV immunofluorescence in other cells of the septa of the connective septum or in the vessels cannot be excluded." means there are only type I, type II cells, nerve profiles and some insignificant amount of other cells, please indicate it clearly. Maybe some other support cells (like blood cells) can be present in your slices and show immunofluorescence.
Author Response
Round 3 – Reviwer 2
Well, I think you've clarified most of it, although I would like to see the answers within the new version.
Unfortunately, most of interested readers, including me, will not be well aware of terminology used in neuronal development biology, such as "nerve profile", so yes, please try to unify the terms throughout your next manuscripts.
The only thing that is left is co-localization with S100 to indicate the presence of TRPV in type II cells. If "The remaining TRPV4 immunofluorescent cells correspond to type II glomic cells, although the presence of TRPV immunofluorescence in other cells of the septa of the connective septum or in the vessels cannot be excluded." means there are only type I, type II cells, nerve profiles and some insignificant amount of other cells, please indicate it clearly. Maybe some other support cells (like blood cells) can be present in your slices and show immunofluorescence.
Once again we thank the Reviewer for his/her constructive criticisms. With his/her help the manuscript has improved considerably since the first version and it can have a larger readership.
We have now answered the questions you have asked us and we will take your recommendations into account for future studies.